# Conversion Therapy to Transplant or Surgical Resection in Patients with Unresectable Hepatocellular Carcinoma Treated with Boosted Dose of Yttrium-90 Radiation Segmentectomy

**DOI:** 10.3390/cancers16173024

**Published:** 2024-08-30

**Authors:** Sam Y. Son, Ruben Geevarghese, Brett Marinelli, Ken Zhao, Anne Covey, Aaron Maxwell, Alice C. Wei, William Jarnagin, Michael D’Angelica, Hooman Yarmohammadi

**Affiliations:** 1Department of Radiology, Interventional Radiology Service, Memorial Sloan Kettering Cancer Center, 1275 York Ave, New York, NY 10065, USA; sam.son.y@outlook.com (S.Y.S.); geevarr1@mskcc.org (R.G.); marinelb@mskcc.org (B.M.); zhaok@mskcc.org (K.Z.); coveya@mskcc.org (A.C.); 2Department of Radiology, Division of Interventional Radiology, Warren Alpert Medical School of Brown University, One Prospect Steet, Providence, RI 02912, USA; aaron_maxwell@brown.edu; 3Department of Surgery, Hepatopancreatobiliary Service, Memorial Sloan Kettering Cancer Center, 1275 York Ave, New York, NY 10065, USA; weia@mskcc.org (A.C.W.); jarnagiw@mskcc.org (W.J.); dangelim@mskcc.org (M.D.)

**Keywords:** hepatocellular carcinoma, yttrium-90, radioembolization, conversion therapy, radiation segmentectomy

## Abstract

**Simple Summary:**

Potential curative options for hepatocellular carcinoma include liver transplant, surgical resection, and ablation. Unfortunately, only 15–30% of patients are eligible for curative options at the time of presentation. Therefore, it is crucial to explore and evaluate treatment methods than can potentially convert nonsurgical candidates into surgical candidates. One such treatment option that can be used as a conversion strategy is yttrium-90 radioembolization. This study examines the success rate of using boosted dose radiation segmentectomy to convert nonsurgical candidates into eligible candidates for transplant or surgical resection.

**Abstract:**

Background/Objectives: The aim of this study was to assess the efficacy of boosted dose yttrium-90 radioembolization (TARE) as a modality for conversion therapy to transplant or surgical resection in patients with unresectable hepatocellular carcinoma (HCC). Methods: In this single-center retrospective study, all patients with a diagnosis of HCC who were treated with boosted dose TARE (>190 Gy) between January 2013 and December 2023 were reviewed. Treatment response and decrease in tumor size were assessed with the RECIST v1.1 and mRECIST criteria. Milan and University of California, San Francisco (UCSF), criteria were used to determine transplant eligibility, and Barcelona Clinic Liver Cancer (BCLC) surgical resection recommendations were used to evaluate tumor resectability. Results: Thirty-eight patients with primary HCC who were treated with boosted dose TARE were retrospectively analyzed. The majority of the patients were Child–Pugh A (*n* = 35; 92.1%), BCLC C (*n* = 17; 44.7%), and ECOG performance status 0 (*n* = 25; 65.8%). The mean sum of the target lesions was 6.0 cm (standard deviation; SD = 4.0). The objective response rate (ORR) was 31.6% by RECIST and 84.2% by mRECIST. The disease control rate (DCR) was 94.7% by both RECIST and mRECIST. Among patients outside of Milan or UCSF, 13/25 (52.0%, Milan) and 9/19 (47.4%, UCSF) patients were successfully converted to within transplant criteria. Of patients who were initially unresectable, conversion was successful in 7/26 (26.9%) patients. Conclusions: This study provides further real-world data demonstrating that boosted-dose TARE is an effective modality for conversion of patients with unresectable HCC to transplant or resection.

## 1. Introduction

Hepatocellular carcinoma (HCC) represents more than 80% of primary liver cancers and is a major cause of cancer deaths globally [1]. Potentially curative treatment options include surgical resection, transplantation, and ablation; however, most patients are outside of the window for curative options due to advanced disease at the time of diagnosis [2]. Current Barcelona Clinic Liver Cancer (BCLC) guidelines recommend systemic treatment, transarterial chemoembolization (TACE), or transarterial radioembolization (TARE) for this patient population, but overall survival (OS) outcomes remain poor compared to curative options [3]. Thus, treatment approaches able to convert patients to within the eligibility criteria for curative therapy are crucial and continue to gain interest within the community. This new concept is referred to as the “treatment stage migration” strategy, in which the treatment option will ultimately allow for the patient to be moved over to another treatment option like surgery in order to obtain a higher success rate [4].

Conversion therapy is a treatment strategy already established in the treatment of solid tumors such as colorectal and gastric cancers and liver metastases that increases opportunities for transplant or surgical resection. In HCC, studies have demonstrated the value of conversion therapy, especially with immune checkpoint inhibitors (ICIs), antiangiogenic agents, and/or locoregional therapies (LRTs) [5,6,7]. Patients with advanced HCC who were converted to resection or transplant experienced significantly prolonged OS according to longitudinal studies, and current investigative efforts are focused on optimizing various conversion therapy strategies [6,8].

Yttrium-90 radioembolization (TARE) is a locoregional modality that offers several advantages when used for conversion therapy. It provides longer time to progression compared to other LRTs and is well tolerated by patients, contributing to quality-of-life benefits [9]. In addition, TARE can induce concurrent tumor shrinkage and contralateral hepatic lobe hypertrophy [10,11]. Delivering ablative doses of TARE in 1 or 2 segments is known as radiation segmentectomy and has been proven to be an effective curative treatment option in certain patient populations [12]. Radiation segmentectomy has been successful in bridging to transplant or downstaging patients to surgery [13,14,15]. 

Utilization of a dose larger than 190 Gy to 1 or 2 segments of the liver is known as a boosted dose of TARE. This study presents the real-world experience of boosted-dose TARE as a modality for converting ineligible patients to within transplant and resection criteria, as performed at a tertiary cancer center.

## 2. Materials and Methods

### 2.1. Study Design 

In this single-center retrospective study, the radiology picture archiving and communication system was initially searched for all patients with a diagnosis of HCC who were treated with TARE between January 2013 and December 2023. Patients previously treated with other locoregional therapies or prior surgery/resection were included. Only patients who received a >190 Gy tumor perfusion liver dose were included. Patient demographics, medical history, tumor characteristics, labs, and treatment history were obtained from the clinical information system (CIS). This study was approved by the Institutional Review Board and was in compliance with the Health Insurance Portability and Accountability Act.

### 2.2. Yttrium-90 Transarterial Radioembolization 

Before treatment, a mapping arteriogram was performed with technetium-99 m-labeled macroaggregated albumin infusion and single photon emission computed tomography (SPECT/CT). The tumor vascular supply, presence of hepatopulmonary shunt, and radiotracer distribution were evaluated during this process. Patients would return for treatment after 2–4 weeks if they were confirmed to be eligible for TARE based on the arteriogram and SPECT/CT. The medical internal radiation dose (MIRD) model was used for the calculation of the Y90 prescription activity [16]. TARE was performed with radial or femoral arterial access, and Y90 microspheres were delivered directly to the tumor arterial supply. The treatment was performed with TheraSphere^®^ Y90-glass microspheres (Boston Scientific, Marlborough, MA, USA). SPECT/CT was carried out post-procedure to assess possible extrahepatic delivery and confirm appropriate tumor coverage.

### 2.3. Eligibility for Transplant and Resection 

Patients were assessed for transplant eligibility according to the Milan and UCSF criteria. The Milan criteria defines transplant eligibility as a single tumor ≤ 5 cm or up to three tumors, none larger than 3 cm each, with no evidence of extrahepatic spread or macrovascular invasion [17]. The UCSF criteria defines eligibility as a single tumor ≤ 6.5 cm or up to three tumors, none larger than 4.5 cm each, with no evidence of extrahepatic spread or macrovascular invasion [18]. Eligibility for resection was assessed according to the BCLC recommendations. Patients were considered technically resectable if they had adequate future liver remnant (FLR) and a single tumor without vascular invasion or extrahepatic spread. Adequate FLR was defined as more than 20% of the total liver volume in non-cirrhotic liver and more than 30% in cirrhotic livers [19]. Baseline radiological assessments of transplant and resection eligibility were carried out for all patients based on the pre-treatment multiphase CT or MRI. Follow-up radiological assessments after TARE were performed at 1 month, 3 months, and every 3 months thereafter.

### 2.4. Evaluation of Treatment Response

The best treatment responses and changes in tumor size per Response Evaluation Criteria in Solid Tumors (RECIST v1.1) and modified RECIST (mRECIST) within 6 months after the initial TARE were used to evaluate the treatment efficacy [20]. The treatment categories were as follows: complete response (CR), partial response (PR), stable disease (SD), and progressive disease (PD). Disease control (DC) was defined as CR, PR, or SD, and objective response (OR) was defined as CR and PR. Local tumor progression (LTP), distant hepatic progression (DHP), and extrahepatic progression (EHP) were assessed until the time of death or last follow-up. Progression in the treated areas of the liver was termed LTP, in untreated areas as DHP, and outside of the liver as EHP.

### 2.5. Statistical Analysis 

Outcomes were reported and summarized with descriptive statistics when appropriate. Categorical variables and continuous variables were compared with Fisher’s exact test and Student’s *t*-test, respectively. Logistic regression analysis was performed to investigate prognostic factors associated with successful conversion to transplant or resection after TARE. All statistical tests were two-tailed and considered statistically significant at *p* < 0.05. Statistical analyses were performed using R statistical computing software (version 4.3.3; R Core Team 2024).

## 3. Results

Thirty-eight patients with HCC who were treated with boosted-dose TARE were retrospectively reviewed and analyzed. The cohort consisted of mostly Child–Pugh A (92.1%), BCLC C (44.7%), and ECOG performance status 0 (65.8%) patients. Multifocal disease was present in 55.3% of the cohort, and the mean largest tumor diameter was 5.2 cm (SD = 2.9). Patients with multifocal disease had disease in a maximum number of two segments. The mean sum of the target lesions was 6.0 cm (SD = 4.0). There were 8/38 (21.1%) patients who had portal vein tumor thrombosis at the time of treatment. The mean perfused liver dose was 349 Gy (SD = 148). The baseline patient and tumor characteristics are summarized in Table 1.

The best treatment response and mean size change from baseline of the target lesions within 6 months per RECIST v1.1 and mRECIST criteria are summarized in Table 2 and Figure 1. The objective response rate (ORR) was 31.6% by RECIST v1.1 and 84.2% by mRECIST. The disease control rate (DCR) was 94.7% by both RECIST v1.1 and mRECIST. There was PD in two patients by both the RECIST v1.1 and mRECIST criteria. Among patients with OR to TARE, the best mean percent change in baseline tumor size within 6 months was −49.1% (SD = 17.6%) by RECIST v1.1 and −76.5% (SD = 21.8%) by mRECIST. 

Additionally, these responders demonstrated a reduction in baseline tumor size of −2.5 cm (SD = 1.3) by RECIST v1.1 and −3.9 cm (SD = 2.1) by mRECIST within 6 months of treatment. 

Prior to TARE, 25 and 19 patients were outside of the Milan and UCSF criteria, respectively. Of these patients, 13/25 (52.0%, Milan) and 9/19 (47.4%, UCSF) patients were successfully converted to within transplant criteria. The mean time to conversion after TARE was 2.4 months (SD = 2.0, Milan) and 2.7 months (SD = 2.3, UCSF). The reasons for conversion after TARE were either shrinkage of the tumors or that the tumor within the vein appeared to be non-viable following imaging. Among patients who were within transplant criteria after TARE, four patients dropped out of Milan and UCSF eligibility due to new portal vein tumor thrombus (*n* = 1), bilobar tumor progression (*n* = 2), and extrahepatic spread (*n* = 1). The mean time to dropout was 13.2 months (SD = 13.6). 

Of patients who were initially unresectable (*n* = 26), conversion to resection eligibility was successful in 7/26 (26.9%) patients. The mean time to conversion after TARE was 3.8 months (SD = 2.4). The reasons for becoming eligible for resection were disappearance of lesions (*n* = 4), decrease in tumor size (*n* = 1), and resolution of viable tumor in the vein (*n* = 2). One patient was successfully treated with a hepatectomy after TARE. Among patients who were anatomically resectable after TARE, four patients dropped out of eligibility due to disease progression with new tumors. The mean time to dropout was 7.2 months (SD = 4.3). Eligibility for transplant and resection before and after TARE is summarized in Table 3. 

After TARE, CR was observed in 31.6% (*n* = 12) of the cohort. Among these patients, 9/12 (75%) had a durable response and did not have any evidence of residual disease during their follow-up. Three patients (25%) had progression of disease after a CR to TARE. The mean time to progression in these patients was 8.2 months (SD = 3.5).

Univariate logistic regression analysis showed that OR per RECIST v1.1 after TARE was a positive prognostic factor (7.11 OR, 1.02–49.5 CI, *p* = 0.047) for successful conversion to resection eligibility. In addition, tumor size > 7 cm was a negative prognostic factor for conversion to Milan (0.08 OR, 0.01–0.86 CI, *p* = 0.035) and UCSF (0.08 OR, 0.01–0.95 CI, *p* = 0.045) criteria. Other baseline patient and tumor characteristics such as age, sex, Child–Pugh score, BCLC, ECOG score, solitary, multifocal, or presence of vascular invasion were not statistically significant.

## 4. Discussion

In this single institutional experience, patients with unresectable HCC were successfully downstaged to transplant (Milan/UCSF) criteria or converted to resection after boosted-dose TARE. This further demonstrates TARE’s role in treating patients with advanced-stage HCC outside the window for resection or transplant. In addition, high doses of Y90 delivered with TARE provided a curative treatment outcome with no evidence of disease for select patients. 

These results are consistent with previous reports on downstaging patients with advanced HCC to surgical resection. DOSISPHERE-01, a multicenter phase II randomized trial that investigated boosted personalized dosimetry (≥205 Gy) vs standard dosimetry (120 ± 20 Gy), reported a conversion to surgery rate of 36% in the personalized dosimetry group compared to 4% in the standard dosimetry group [14]. In the current study, conversion to resection was observed in 26.9% of patients, which was better than the reported standard dosimetry group and lower than the personalized dosimetry group. In the current study, the MIRD model was used for dose calculation, and therefore tumor-absorbed dose was not calculated. Hence, these results support the use of personalized dosimetry and higher mean tumor-absorbed doses. Similar to the current study, Tzedakis et al. and Inarrairaegui et al. reported similar conversion rates, further supporting the use of TARE in conversion therapy to resection [21,22]. 

In the current study, boosted-dose TARE, >190 Gy, was effective as a modality for downstaging patients to transplant criteria. Recent studies including the LEGACY trial have documented higher selective treatment-absorbed doses for radiation segmentectomy [23,24]. Recent studies have demonstrated that absorbed doses exceeding 400 Gy are associated with complete pathologic necrosis and a lower recurrence rate in patients who are bridging to transplant [25,26]. The present study had a conversion rate to transplant criteria in 52.0% and 47.4% of patients based on the Milan and UCSF criteria, respectively. Patients with successful conversion after TARE had a decrease in tumor burden and resolution of enhancement of tumor within the veins. In comparison, Gabr et al. reported a similar rate of downstaging to transplant of 47% in their cohort, consisting of mostly BCLC A/B patients [27]. The median tumor-absorbed dose in Gabr et al.’s study was 260 Gy and 124 Gy for segmental and lobar delivery, respectively [27]. Higher conversion rates were reported in cohorts with earlier-stage HCC at 57% and 80.6% after TARE [28,29]. 

Several patients within transplant and resection criteria after TARE became ineligible during the study’s follow-up period. For transplant candidates, the mean time to ineligibility was 13.2 months, which signifies that TARE can be an effective modality to control disease and bridge patients to transplant while they are on the waitlist. This is consistent with large retrospective studies reporting greater than 95% of patients remaining on the transplant list using TARE as the locoregional option [25,27]. 

As with radiation lobectomy, boosted-dose TARE can also serve as a biologic test-of-time to assess tumor aggressiveness, allowing for careful patient selection and risk stratification for surgical interventions. For resection candidates, the mean time to ineligibility was 7.2 months, indicating that there is a narrow window of opportunity for surgical interventions. In a systemic review and meta-analysis on patients with HCC that were surgically resected after TARE, Khan et al. reported 16 studies on 276 patients [30]. The median time to resection after TARE ranged from 2 to 12.5 months. This is similar to the current study. Khan et al. concluded that liver resection after TARE was safe in well-selected patients with a 30-day mortality rate of zero [30]. 

The limitations of this study are the potential for bias due to its retrospective design, small sample size, and single-institutional data. Additionally, the institution of the present study is not a liver transplant center and lacks complete data on candidates who ultimately underwent transplantation at other tertiary centers. 

## 5. Conclusions

The emerging evidence for converting patients to transplant and resection with locoregional and systemic therapies is promising. This study provides further real-world data demonstrating that boosted-dose TARE is an effective modality for downstaging to transplant and conversion to resection. Recent evidence strongly supports the use of personalized dosimetry and administering higher mean tumor-absorbed doses. 

## Figures and Tables

**Figure 1 cancers-16-03024-f001:**
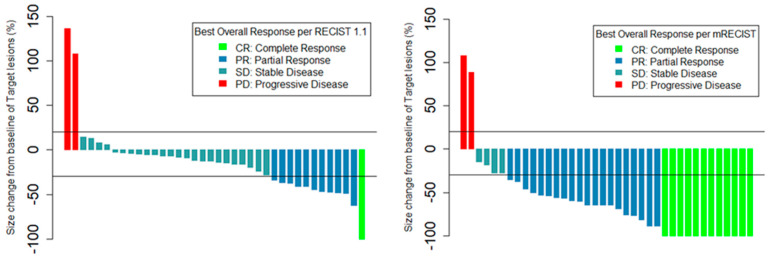
Best percent change in size of target lesions per RECIST v1.1 and mRECIST.

**Table 1 cancers-16-03024-t001:** Baseline patient and tumor characteristics.

	*n* = 38
**Mean age ± SD, years**	70.8 ± 10.9
**Male gender, No. (%)**	32 (84.2)
**Race/ethnicity, No. (%)**	
White	22 (57.9)
Asian	11 (28.9)
Black	2 (5.3)
Hispanic	3 (7.9)
**Etiology, No. (%)**	
HCV	8 (21.1)
HBV	8 (21.1)
Alcohol	4 (10.5)
NAFLD	3 (7.9)
Multiple	3 (7.9)
Unknown	11 (28.9)
Other	1 (2.6)
**Child–Pugh, No. (%)**	
A (score 5–6)	35 (92.1)
B (score 7–9)	3 (7.9)
**BCLC classifications, No. (%)**	
A	11 (28.9)
B	10 (26.3)
C	17 (44.7)
**ECOG grade, No. (%)**	
0	25 (65.8)
1	8 (21.1)
2	5 (13.2)
**AFP, No. (%)**	
≥200 ng/mL	8 (21.1)
<200 ng/mL	25 (65.8)
n/a	5 (13.2)
**Tumor characteristics, No. (%)**	
Solitary	17 (44.7)
Multifocal	21 (55.3)
Mean largest tumor diameter ± SD, cm	5.2 ± 2.9
Median largest tumor diameter, cm (range)	4.3 (1.5–15.1)
Mean sum of target lesions, cm	6.0 ± 4.0
Median sum of target lesions, cm (range)	4.5 (1.5–17.6)
Portal vein tumor thrombosis, No. (%)	8 (21.1)
**Prior treatment history**	
Surgical resection, No.	4
Systemic therapy, No.	2
TAE, No.	7
Ablation, No.	2

HCV—hepatitis C virus, HBV—hepatitis B virus, NAFLD—non-alcoholic fatty liver disease, BCLC—Barcelona Clinical Liver Cancer, ECOG—Eastern Cooperative Oncology Group, AFP—alpha fetoprotein, TAE—transarterial embolization.

**Table 2 cancers-16-03024-t002:** Best treatment response of target lesions within 6 months.

	RECIST v1.1		mRECIST	
	*n* (%)	Mean Size Change in Target Lesions from Baseline ± SD	*n* (%)	Mean Size Change in Target Lesions from Baseline ± SD
Complete response	1 (2.6)	−100%	12 (31.6)	−100%
Partial response	11 (28.9)	−44.4 ± 7.7%	20 (52.6)	−62.4 ± 14.9%
Stable disease	24 (63.2)	−7.6 ± 10.6%	4 (10.5)	−21.9 ± 6.6%
Progressive disease	2 (5.3)	+122.3 ± 19.8%	2 (5.3)	+98.5 ± 13.9%
Objective response	12 (31.6)	−49.1 ± 17.6%	32 (84.2)	−76.5 ± 21.8%
Disease control	36 (94.7)	−21.4 ± 23.7%	36 (94.7)	−70.5 ± 27.0%

**Table 3 cancers-16-03024-t003:** Summary of surgical resection and transplant eligibility before and after TARE.

	Pre-TARE	Post-TARE	Successful Conversion after TARE	Mean Time to Conversion after TARE ± SD	Eligibility Dropout
Milan Criteria	13/38 (34.2%)	26/38 (68.4%)	13/25 (52.0%)	2.4 ± 2.0 months	4/26 (15.4%)
UCSF Criteria	19/38 (50.0%)	28/38 (73.7%)	9/19 (47.4%)	2.7 ± 2.3 months	4/28 (14.3%)
Resectable	12/38 (31.6%)	19/38 (50.0%)	7/26 (26.9%)	3.8 ± 2.4 months	4/19 (21.1)

## Data Availability

The data presented in this study are available on request from the corresponding author due to institutional restrictions.

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
