# Peer review of "Conversion Therapy to Transplant or Surgical Resection in Patients with Unresectable Hepatocellular Carcinoma Treated with Boosted Dose of Yttrium-90 Radiation Segmentectomy"

_cancers, 2024, doi:10.3390/cancers16173024_

Round 1

Reviewer 1 Report

Comments and Suggestions for Authors

The article “Conversion therapy to transplant or surgical resection in patients with unresectable hepatocellular carcinoma treated with a boosted dose of yttrium-90 radiation segmentectomy” describes a retrospective, single-center study, focused on 38 HCC patients.

Due to the materials and methods only referring to a boosted dose of > 190 Gy, the article's dose definition method is not adequately developed.  It is necessary to identify the software used for dosimetric estimation and the method used for dose calculation.

In addition, resin or glass microsphere subgroups could be taken into consideration.

Indeed, the analysis pools here the 2 types of microspheres. It turns out that the dosimetric objectives are completely different depending on the type of microspheres.

For example in the context of a segmentectomy in “single compartment analysis” the dose to the perfused volume must be > 400 Gy for glass microspheres while the objective is > 150 Gy for resin microspheres.

Each type of microsphere should have a detailed report on the mean perfused liver dose instead of using a global mean perfused liver dose of 349 Gy.

The population could be more precisely described in the case of a multifocal tumor to accurately determine the targeted volume.

One of the main points of SIRT is based on the dosimetric approach of treatment. It could be interesting to quote the main recommendations on this subject are not cited in this article (i.e. Weber et al, EJNMMI (2022) 49 : 1682-1699 ; Salem et al, EJNMMI (2023) 50:328-343).

Author Response

Comment 1: The article “Conversion therapy to transplant or surgical resection in patients with unresectable hepatocellular carcinoma treated with a boosted dose of yttrium-90 radiation segmentectomy” describes a retrospective, single-center study, focused on 38 HCC patients.

Due to the materials and methods only referring to a boosted dose of > 190 Gy, the article's dose definition method is not adequately developed.  It is necessary to identify the software used for dosimetric estimation and the method used for dose calculation.

Response 1: Thank you for your comment. The dosimetry method used in all patients was Medical Internal Radiation Dose (MIRD) model. This has been described in the method section line 94. The following reference was used and is now added to the manuscript:

Lewandowski RJ, Gabr A, Abouchaleh N, Ali R, Al Asadi A, Mora RA, Kulik L, Ganger D, Desai K, Thornburg B, Mouli S, Hickey R, Caicedo JC, Abecassis M, Riaz A, Salem R. Radiation Segmentectomy: Potential Curative Therapy for Early Hepatocellular Carcinoma. Radiology. 2018 Jun;287(3):1050-1058. doi: 10.1148/radiol.2018171768. Epub 2018 Apr 24. PMID: 29688155.

Comment 2: 

In addition, resin or glass microsphere subgroups could be taken into consideration.

Indeed, the analysis pools here the 2 types of microspheres. It turns out that the dosimetric objectives are completely different depending on the type of microspheres.

For example in the context of a segmentectomy in “single compartment analysis” the dose to the perfused volume must be > 400 Gy for glass microspheres while the objective is > 150 Gy for resin microspheres.

Each type of microsphere should have a detailed report on the mean perfused liver dose instead of using a global mean perfused liver dose of 349 Gy.

Response 2: We agree with the reviewer. In the current study, only Therasphere was used. The method section mentions that either Therasphere or Sirsphere were used. However, by this comment we meant that both products were available to the IR doctors. However, all patients in the study were treated with Therasphere. We revised the method section and clarified it in line 97.

Comment 3: The population could be more precisely described in the case of a multifocal tumor to accurately determine the targeted volume.

Response: We added some more detail to the manuscript. (Line 137)

Comment 4: One of the main points of SIRT is based on the dosimetric approach of treatment. It could be interesting to quote the main recommendations on this subject are not cited in this article (i.e. Weber et al, EJNMMI (2022) 49 : 1682-1699 ; Salem et al, EJNMMI (2023) 50:328-343).

Response 4: Both references were included in the discussion. (line 217).

Reviewer 2 Report

Comments and Suggestions for Authors

Congratulations for the authors in writing this paper in an interesting area of treatment of liver malignancy.

The quality of paper is good.The study design is appropriate to answer the research and the study my repeated. The presentetion of the work is good however it is not  clear in Table 1. Baseline Patient and Tumor Characteristics, how  among 38 patients treated with boosted dose TARE, 11 (28.9 %) were in group A- BCLC classifications. I am not agree with the selection of patients.  Group A patients in BCLC  model don't require  TARE treatment, they are in themselves eligible for resection, trasplantation or ablation.Why they were treated with TARE?

Author Response

The quality of paper is good.The study design is appropriate to answer the research and the study my repeated. The presentetion of the work is good however it is not  clear in Table 1. Baseline Patient and Tumor Characteristics, how  among 38 patients treated with boosted dose TARE, 11 (28.9 %) were in group A- BCLC classifications. I am not agree with the selection of patients.  Group A patients in BCLC  model don't require  TARE treatment, they are in themselves eligible for resection, trasplantation or ablation. Why they were treated with TARE?

Respond: These patients were not surgical candidates due to comorbidities making surgical resection or transplant high risk for complication.